# Patient-reported pain and physical health for acupuncture and chiropractic care delivered by Veterans Affairs versus community providers

**Claire E. O'Hanlon**[1,2], **Steven B. Zeliadt**[3,4], **Rian DeFaccio**[3], **Lauren Gaj**[5], **Barbara G. Bokhour**[5,6], **Stephanie L. Taylor**[1,7,8]*

1 Veterans Affairs Greater Los Angeles Healthcare System, Center for the Study of Healthcare Innovation, Implementation & Policy (CSHIIP), Los Angeles, California, United States of America, 2 RAND Corporation, Santa Monica, California, United States of America, 3 Veterans Affairs Puget Sound Health Care System, VA Center of Innovation for Veteran-Centered and Value-Driven Care, Seattle, Washington, United States of America, 4 Department of Health Systems and Population Health, Hans Rosling Center for Population Health, University of Washington School of Public Health, Seattle, Washington, United States of America, 5 VA Bedford Healthcare System, Center for Healthcare Organization and Implementation Research (CHOIR), Bedford, Massachusetts, United States of America, 6 Division of Preventive and Behavioral Medicine, Department of Population and Quantitative Health Sciences, University of Massachusetts Chan Medical School, Worcester, Massachusetts, United States of America, 7 Department of Health Policy and Management, Fielding School of Public Health, Los Angeles, California, United States of America, 8 Department of Medicine, David Geffen School of Medicine at UCLA, Los Angeles, California, United States of America

* stephanie.taylor8@va.gov

## Abstract

### Background

Acupuncture and chiropractic care are evidence-based pain management alternatives to opioids. The Veterans Health Administration (VA) provides this care in some VA facilities, but also refers patients to community providers. We aimed to determine if patient-reported outcomes differ for acupuncture and chiropractic care from VA versus community providers.

### Materials and methods

We conducted an observational study using survey outcome data and electronic medical record utilization data for acupuncture and chiropractic care provided in 18 VA facilities or in community facilities reimbursed by VA. Study participants were users of VA primary care, mental health, pain clinic, complementary and integrative therapies, coaching or education services in 2018–2019. Patients received 1) 4+ acupuncture visits (N = 201) or 4+ chiropractic care visits (N = 178) from a VA or community provider from 60 days prior to baseline to six-months survey and 2) no acupuncture or chiropractic visits from 1 year to 60 days prior to baseline. Outcomes measured included patient-reported pain (PEG) and physical health (PROMIS) at baseline and six-month surveys. Multivariate analyses examined outcomes at six months, adjusting for baseline outcomes and demographics.

**Data Availability Statement:** The United States Department of Veterans Affairs (VA) places legal restrictions on access to veteran's health care data,

which includes both identifiable and de-identified data, and sensitive patient information. The analytic data sets used for this project are not permitted to leave the VA firewall without a Data Use Agreement (DUA). This limitation is consistent with other studies based on VA data. However, VA data are made freely available to investigators behind the VA firewall with an approved VA study protocol. Programming code is available in the form of Supporting information files uploaded alongside this manuscript. For more information about data access within VA, please visit https://www.virec. research.va.gov or contact the VA Information Resource Center (VIReC) at VIReC@va.gov.

**Funding:** This evaluation was funded as a quality improvement project by the Office of Patient Centered Care and Cultural Transformation and VA QUERI program (PEC 13-001, PI: BGB), https:// www.queri.research.va.gov/. The funders had no role in study design, data collection and analysis, decision to publish, or preparation of the manuscript.

**Competing interests:** The authors have declared that no competing interests exist.

## Results

In unadjusted analyses, pain and physical health improved for patients receiving community-based acupuncture, while VA-based acupuncture patients experienced no change. Unadjusted analyses also showed improvements in physical health, but not pain, for patients receiving VA-based chiropractic care, with no changes for community-based chiropractic care patients. Using multivariate models, VA-based acupuncture was no different from community-based acupuncture for pain (-0.258, p = 0.172) or physical health (0.539, p = 0.399). Similarly, there were no differences between VA- and community-based chiropractic care in pain (-0.273, p = 0.154) or physical health (0.793, p = 0.191).

## Conclusions

Acupuncture and chiropractic care were associated with modest improvements at six months, with no meaningful differences between VA and community providers. The choice to receive care from VA or community providers could be based on factors other than quality, like cost or convenience.

## Introduction

Whether it is better to invest in providing health care in-house versus contracting with outside providers is a question of considerable policy interest to the Veterans Health Administration (VA). VA is the largest integrated health care system in the United States, with over 367,200 employees in 1,293 facilities serving 9 million Veterans annually [1]. Legislation over the last decade such as the MISSION Act [2] aims to improve access to care by making it easier for Veterans to receive VA-financed care from community (non-VA) providers. As this access expands, clinicians need to know if there are advantages or disadvantages of referring patients to one care setting over another (e.g., care coordination issues [3], effectiveness), Veterans need to know where they can obtain convenient, high-quality care.

Veterans experience chronic pain [4] at higher rates than the general population. Although Veterans experience similarly concerning rates of opioid use disorder compared to civilians [5], Veterans wounded in combat have especially high rates of prescription opioid and sedative misuse [6]. As such, VA has prioritized Veterans' access to evidence-based non-pharmacological pain management [7] including complementary and integrative medicine modalities [8] including acupuncture [9–12] and chiropractic care [13–16]. Many VA facilities provide this care in-house from dedicated providers, though use and availability vary widely [17, 18]. VA often refers patients to community providers for acupuncture and chiropractic care when VA-based services are not available or easily accessible [19, 20].

Comparing the quality of care provided by VA and community providers has become a research priority as administrators and policymakers make decisions about whether to invest in additional VA providers and services or outsource care [21]. VA may be best suited to provide high quality care to Veteran patients because Veterans constitute a complex and unique patient population with health care challenges resulting from the exposures and experiences of military service [22–24]. Numerous studies comparing the quality of care provided by VA and community settings found that VA care almost always performs similar to or better than care provided in the community [25–28]. To our knowledge, no other studies have examined differences in outcomes of complementary and integrative therapies for Veteran patients receiving care from VA- and community-based providers. In this study, we aim to determine if

there are differences in patient-reported outcomes in VA and community acupuncture and chiropractic care for Veteran patients.

## Methods

### Research ethics

The project generating these findings was conceived and conducted as a non-research operations activity conducted as part of a congressionally-mandated internal operational assessment of VHA's Whole Health pilot program included in the Comprehensive Addiction and Recovery Act (CARA) of 2016 (Public Law No:114–198). The results from this evaluation were derived from this non-research operations activity in accordance with VHA Handbook 1058.05 and Program Guide 1200.21 and are therefore exempt from review by Veterans Affairs Institutional Review Board and informed consent procedures.

### Survey sampling approach

Patient-reported demographic and health outcomes were obtained from two mailed paper surveys. Veteran patients were selected from one of 18 facilities (one facility in every Veterans Integrated Service Network [VISN], VA's regional health systems) that had been previously selected to participate in VA's Whole Health demonstration project [29]. Our sample comprised patients with a recent primary care, mental health, or pain clinic visit who had chronic musculoskeletal pain diagnoses at the time of the visit or recent use of Whole Health services, which include a variety of complementary and integrative therapies, coaching or education. Several waves of surveys were distributed and collected between March 2018 and January 2020. Survey details have been previously reported [30].

### Survey patient-reported outcomes

The survey comprised 22 measures of patient-reported outcomes. Two patient-reported outcomes, pain and physical health, are included in this analysis. Pain was assessed using the mean score of the 3-item Pain, Enjoyment of Life, and General Activity (PEG) scale [31]. The score is the average of the three items (range: 0–10). Physical health was assessed using four individual questionnaire items from the PROMIS-10, assessing overall physical health, physical activities, fatigue, and pain [32]. The score is the sum of all four components converted to a T-score relative to national averages (range: 16–68).

### Administrative data on utilization

Administrative data on acupuncture and chiropractic care utilization provided by VA and community providers reimbursed by VA and demographic data not available in the survey were obtained from the VA Corporate Data Warehouse on October 8, 2021. We calculated the number of acupuncture (traditional or body acupuncture only; auricular or "battlefield" acupuncture visits were excluded) and chiropractic care visits made to VA and community providers during the "study period," (i.e., 60 days prior to baseline survey completion to six-month survey completion). Once survey responses and administrative data were linked, data were deidentified and the authors no longer had access to information that could identify individual participants.

### Study sample

Our sample was composed of Veterans who completed baseline and six-month surveys and received a "dose" of acupuncture or chiropractic care provided or paid for by VA during the

study period. We defined dose as four or more visits based on expert opinion that a smaller number of visits over 6 months would be unlikely to have a sustained effect on pain or physical health. As we were comparing baseline and six-month follow-up survey outcomes, we wanted to examine new users (those not using these therapies at baseline), so we excluded patients with any acupuncture or chiropractic care visits from one year prior to 60 days prior to the baseline survey (start of the "study period").

A total of 6,853 patients had complete baseline and six-month surveys (Fig 1). We excluded from the acupuncture analyses the 6,228 patients receiving no acupuncture visits during the study period, 219 patients receiving 1–3 acupuncture visits during the study period, and 182 patients receiving any acupuncture visits from one year prior to 60 days prior to the baseline survey. We also excluded from the chiropractic care analysis the 6,251 patients receiving no chiropractic care visits during the study period, 198 patients receiving 1–3 chiropractic care visits during the study period, and 204 patients receiving any chiropractic care visits from one year prior to 60 days prior to the baseline survey. Additionally, 23 patients receiving acupuncture and 22 patients receiving chiropractic care from both VA and community providers during the study period were excluded. This resulted in an analytic sample of 201 patients receiving acupuncture (109 in the VA and 92 in the community) and an analytic sample 178 patients receiving chiropractic care (110 in the VA and 68 in the community).

## Analysis

We first analyzed the sample to determine if there were any differences in demographic characteristics using paired t-tests or chi-square tests as appropriate. Differences in observed health outcomes scores between baseline and six months were assessed using paired t-tests. We then used linear regression models to examine differences in outcomes at six months, comparing VA users to community care users [33]. We conducted a simple bivariate analysis controlling only for baseline score. We then conducted multivariate analysis using linear regression models controlling for baseline score as well as age, sex, race/ethnicity, marital status, VA copay status, educational attainment, region, urbanicity, and driving distance to the nearest VA primary care site (categorized as shown in Table 1). Regressions were of the form:

$$outcome_{6mo} = VAuser(0\ or\ 1) + outcome_{baseline}[+ covariates] \tag{1}$$

We conducted sensitivity analyses by also adjusting for the total number of visits during the study period to account for a potential dose-response effect.

## Results

### Acupuncture

The 109 patients receiving VA acupuncture included in this study had a mean of 6.9 visits (median: 6) during the study period (range: 4–16, interquartile range [IQR]: 5–9), while the 92 community acupuncture patients had a mean of 11.0 visits (median: 10; range: 4–28; IQR: 7–13). The VA and community acupuncture patients differed on age and region of residence (Table 1).

In unadjusted analyses, Veterans using community providers for acupuncture had improvements in pain and physical health at six months compared to baseline assessments (Table 2). However, those using VA providers reported similar scores on both outcomes at

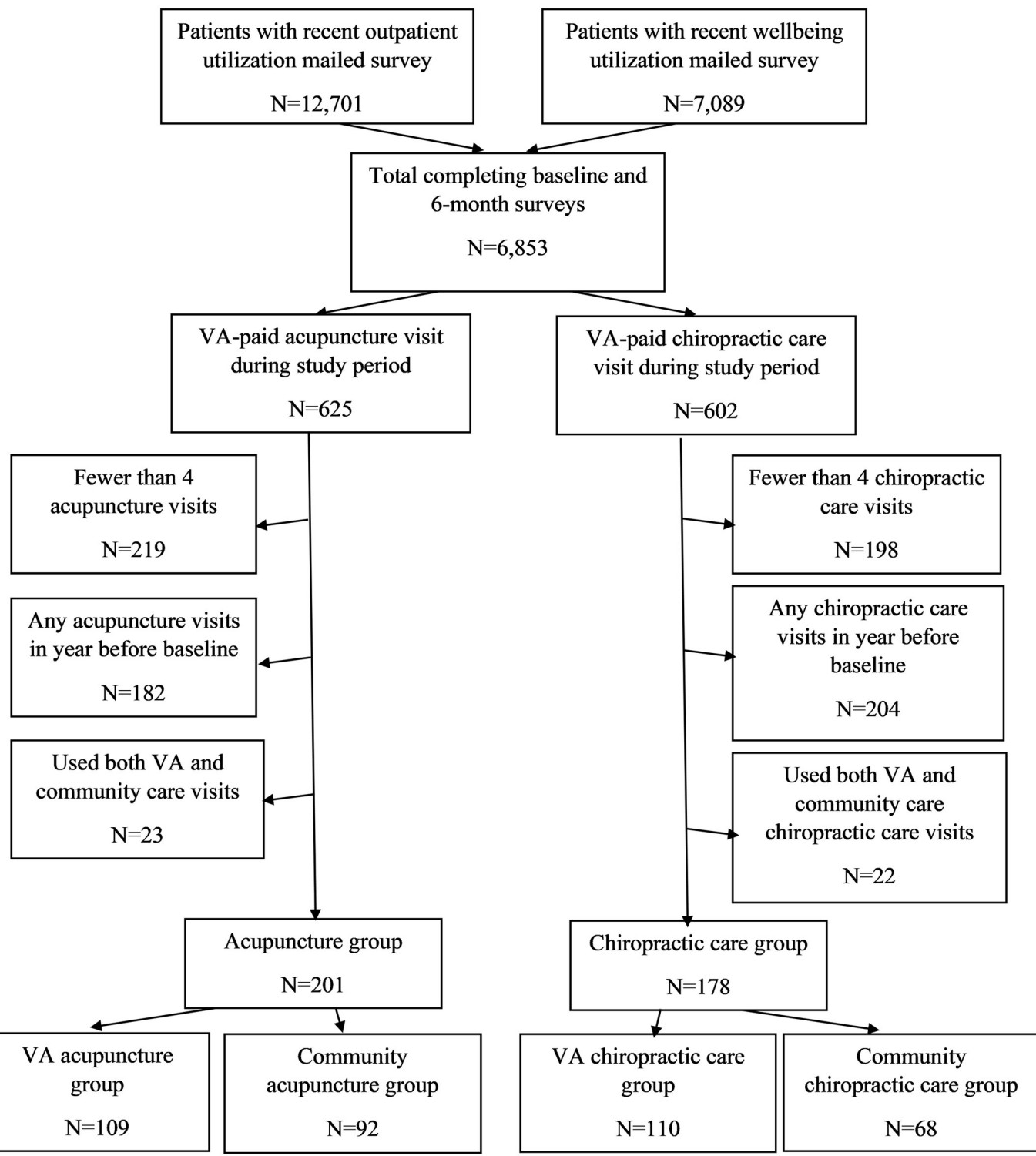

**Fig 1. Patient flow diagram.**

**Table 1. Demographic characteristics of sample.**

| | Acupuncture (N = 201) | | | Chiropractic (N = 178) | | |
|---|---|---|---|---|---|---|
| | VA users | Community users | p | VA users | Community users | p |
| N patients | 109 | 92 | | 110 | 68 | |
| Age (years) | 64.9 | 61.3 | 0.035 | 61.2 | 60.2 | 0.596 |
| *Sex* | | | 0.733 | | | 0.348 |
| Male | 80.7% | 82.6% | | 86.4% | 80.9% | |
| Female | 19.3% | 17.4% | | 13.6% | 19.1% | |
| *Race* | | | 0.184 | | | 0.423 |
| White/Caucasian | 82.6% | 75.0% | | 71.8% | 72.1% | |
| Black/African-American | 12.8% | 17.5% | | 19.1% | 16.2% | |
| Hispanic/Latino (of any race) | 0.9% | 5.4% | | 5.5% | 2.9% | |
| Other/Missing | 3.7% | 2.2% | | 3.6% | 8.8% | |
| *Marital Status* | | | 0.984 | | | 0.382 |
| Married/Engaged/Partnered | 68.8% | 69.6% | | 68.2% | 63.2% | |
| Unmarried/Divorced/Widowed | 30.3% | 29.3% | | 31.8% | 35.3% | |
| Missing | 0.9% | 1.1% | | - | 1.4% | |
| *VA Copay Status* | | | 0.715 | | | 0.363 |
| Copay required | 7.3% | 9.8% | | 05.5% | 10.3% | |
| No copay required | 92.7% | 90.2% | | 94.5% | 89.7% | |
| *Education* | | | 0.563 | | | 0.479 |
| Some college or less | 58.7% | 63.0% | | 58.2% | 64.7% | |
| 4-year college degree or more | 40.4% | 37.0% | | 41.8% | 35.3% | |
| Missing | 0.9% | 0% | | 0% | 0% | |
| *Location* | | | <0.001 | | | <0.001 |
| East North Central | 2.8% | 13.0% | | 7.2% | 23.5% | |
| East South Central | 4.6% | 9.7% | | 27.3% | 14.7% | |
| Mid-Atlantic | 19.2% | 1.1% | | 2.7% | 4.4% | |
| Mountain | 6.4% | 6.5% | | 10.9% | 19.1% | |
| Northeast | 9.1% | 2.2% | | 0% | 7.4% | |
| Pacific | 1.8% | 15.2% | | 4.5% | 7.4% | |
| South Atlantic | 11.0% | 37.0% | | 15.4% | 7.4% | |
| West North Central | 33.9% | 14.1% | | 27.3% | 11.8% | |
| West South Central | 11.0% | 1.1% | | 4.5% | 4.4% | |
| *Urbanicity* | | | | | | |
| Urban/Suburban | 82.6% | 73.9% | 0.142 | 82.7% | 64.7% | 0.010 |
| Rural | 17.4% | 26.1% | | 17.3% | 35.3% | |
| Nearest VA primary care site (mi) | 14.5 | 16.3 | 0.354 | 13.9 | 17.8 | 0.038 |

p-values reflect significance of t-tests for binary or continuous variables and Chi-squared test for categorical variables. Bolded values indicate significance. Groups may not add up to 100% due to rounding.

baseline and six months. When comparing six-month outcomes for VA to community acupuncture using linear regression models controlling for baseline outcome scores, no differences were observed by provider type (Table 3). These results held when we controlled for the number of visits (S1 Table), which is notable since patients using acupuncture from community providers received more visits than patients using acupuncture from VA providers.

**Table 2. Observed scores at baseline and six months.** Lower pain scores and higher physical health scores are better.

|  | Care Provider | Pain (range 0–10) | | | Physical Health (range 16–68) | | |
|---|---|---|---|---|---|---|---|
|  |  | Baseline score | 6-mo score | p | Baseline score | 6-mo score | p |
| Acupuncture | VA | 6.66 | 6.42 | 0.157 | 36.35 | 37.11 | 0.160 |
|  | Community | 7.13 | 6.74 | 0.013 | 34.37 | 35.91 | 0.014 |
| Chiropractic | VA | 6.55 | 6.26 | 0.085 | 36.97 | 37.82 | 0.026 |
|  | Community | 6.79 | 6.88 | 0.660 | 36.53 | 36.32 | 0.654 |

## Chiropractic care

The 110 VA chiropractic care patients included in this study had a mean of 7.4 visits (median: 6) on average during the study period (range: 4–17, IQR: 5–9), while the 68 community chiropractic care patients had a mean of 10.1 visits (median: 9.5; range: 4–25; IQR: 6.75–12). VA and community chiropractic care patients differed on their region of residence, urbanicity, and distance to the nearest VA primary care clinic (Table 1).

In unadjusted analyses, Veterans using VA providers for chiropractic care had improved physical health at six months compared to baseline but had no improvements in pain (Table 2). Patients using community providers for chiropractic care had no improvements in either pain or physical health.

When comparing VA-based chiropractic care to community chiropractic care using linear regression models controlling for baseline outcome scores, no differences were observed (Table 3). These results held when we controlled for the number of visits (S1 Table), which is notable since patients using chiropractic care received more visits from community providers than VA providers.

## Discussion

This study is the first known comparison of Veterans' patient-reported outcomes of acupuncture and chiropractic care from VA and community-based providers. While we observed some changes at six months with decreased pain and improvement in physical health after using acupuncture and chiropractic care, these improvements were small in magnitude and only statistically significant for Veterans who used acupuncture from community providers, and Veterans who used chiropractic care from VA providers. Because these patients used these services at least four times over the six-month period, it is likely that many of these patients continued to experience burdens associated with chronic pain throughout the study period.

Overall, we did not see meaningful differences between VA-based acupuncture and chiropractic care compared to community providers in our adjusted models. Many things may

**Table 3. Regression coefficients of six-month outcomes associated with care from VA providers compared to care from community providers.**

|  |  | Coefficient[a] (VA relative to community) | p | Coefficient[b] (VA relative to community) | p |
|---|---|---|---|---|---|
| Acupuncture | Pain | -0.152 | 0.316 | -0.258 | 0.172 |
|  | Physical health | 0.766 | 0.137 | 0.539 | 0.399 |
| Chiropractic | Pain | -0.255 | 0.140 | -0.273 | 0.154 |
|  | Physical health | 0.776 | 0.150 | 0.793 | 0.191 |

[a]Controlling for baseline outcome score only
[b]Controlling for baseline outcome score, demographics

influence whether a patient experiences improvements with these therapies [34, 35], but whether VA provides or pays for the care from community providers seems not to be one of them, at least over the six-month period we examined. Our findings, that there were no differences in outcomes of acupuncture and chiropractic care between VA providers and community providers, are important. While most studies comparing VA and non-VA care examine inpatient and emergency care, prior studies of outpatient care have found that VA outpatient providers provide better or similar patient experiences to community outpatient providers [36, 37]. As only a few prior studies have been done comparing processes or outcomes in VA and non-VA outpatient care [38–40], it is notable that acupuncture and chiropractic care outcomes were found to be similar among patients receiving care from VA and community providers, at least with respect to outcomes of care in the real world. It is especially notable that differences in outcomes were not observed while utilization was higher for patients using community providers, which could mean that VA care is more efficient if the cost per visit is the same. There may be other distal benefits to having such care provided within VA, such as coordination and linkages with other kinds of care [18, 41]. The use of such care within or outside the VA could also potentially result in different care utilization cascades downstream, such as for imaging, referrals to specialists, or prescriptions for opioids, which has important implications for costs and outcomes.

There are several important limitations of this study. Our sample was small, and our ability to interpret the modest improvements we observed is limited. As this study did not include a comparison group of patients who did not use these services, we do not know how the observed small magnitude of improvement at six months compares with not having used these therapies. Observed improvements could be due to regression to the mean, or alternatively could represent relatively large improvements if patients would have declined significantly in the absence of using these therapies. We also cannot rule out the potential role of confounding factors in these results. We adjusted for geographic differences in access, demographics, and differences in pain and physical health at baseline. However, these factors may not have controlled for unobservable characteristics influencing patient-reported outcomes and the receipt of care from VA or community providers, as the choice of provider may correlate to other important demographic or health factors. We also did not control for other complementary and integrative health modalities that patients may have been utilizing in addition to acupuncture or chiropractic care, nor could we measure acupuncture or chiropractic care that patients were using but did not seek reimbursement for from VA. Lastly, these results may or may not generalize to the population of Veterans with chronic pain, as this study only includes Veterans who received acupuncture or chiropractic care and were patients at the VA facilities that were part of the VA demonstration project.

## Conclusions

The MISSION Act has made it even easier for Veterans to access care from community providers when care options within VA are not available in a timely manner or within a reasonable travel distance [42]. As outcomes do not seem to greatly vary for new acupuncture and chiropractic care users when receiving care from VA or community providers, this decision could be made on other factors, such as patient preferences, convenience, or cost.

## Supporting information

**S1 Table. Regression coefficients of six-month outcomes associated with care from VA providers compared to care from community providers, controlling for total number of visits.**
(DOCX)

## Author Contributions

**Conceptualization:** Steven B. Zeliadt, Barbara G. Bokhour, Stephanie L. Taylor.

**Data curation:** Claire E. O'Hanlon.

**Formal analysis:** Claire E. O'Hanlon.

**Funding acquisition:** Barbara G. Bokhour, Stephanie L. Taylor.

**Investigation:** Lauren Gaj, Barbara G. Bokhour.

**Methodology:** Rian DeFaccio.

**Resources:** Barbara G. Bokhour.

**Software:** Rian DeFaccio.

**Supervision:** Steven B. Zeliadt, Stephanie L. Taylor.

**Validation:** Rian DeFaccio.

**Visualization:** Claire E. O'Hanlon.

**Writing – original draft:** Claire E. O'Hanlon, Stephanie L. Taylor.

**Writing – review & editing:** Steven B. Zeliadt, Rian DeFaccio, Lauren Gaj, Barbara G. Bokhour.

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
