## [Decision Letter · Decision Letter 0]

5 Mar 2024

PONE-D-24-02440Patient-Reported Pain and Physical Health for Acupuncture and Chiropractic Care Delivered by Veterans Affairs Versus Community ProvidersPLOS ONE

Dear Dr. O'Hanlon,

Thank you for submitting your manuscript to PLOS ONE. After careful consideration, we feel that it has merit but does not fully meet PLOS ONE’s publication criteria as it currently stands. Therefore, we invite you to submit a revised version of the manuscript that addresses the points raised during the review process.

Thank you for your submission, both reviewers have commented that the work is well-written and will be a valuable contribution to the literature. The Reviewers have also provided suggestions for improving the work. Reviewer 2 has provided some additional citations, you may wish to consider these for inclusion but you are not required to include them.

We look forward to receiving your revised manuscript.

Kind regards,

Jenny Wilkinson, PhD

Academic Editor

PLOS ONE

2. In the online submission form you indicate that your data is not available for proprietary reasons and have provided a contact point for accessing this data. Please note that your current contact point is a co-author on this manuscript. According to our Data Policy, the contact point must not be an author on the manuscript and must be an institutional contact, ideally not an individual. Please revise your data statement to a non-author institutional point of contact, such as a data access or ethics committee, and send this to us via return email. Please also include contact information for the third party organization, and please include the full citation of where the data can be found.

3. We notice that your supplementary table (Appendix Table) is included in the manuscript file. Please remove them and upload them with the file type 'Supporting Information'. Please ensure that each Supporting Information file has a legend listed in the manuscript after the references list.

Reviewers' comments:

Reviewer's Responses to Questions

**Comments to the Author**

1. Is the manuscript technically sound, and do the data support the conclusions?

Reviewer #1: Yes

Reviewer #2: Yes

2. Has the statistical analysis been performed appropriately and rigorously? 

Reviewer #1: Yes

Reviewer #2: Yes

3. Have the authors made all data underlying the findings in their manuscript fully available?

Reviewer #1: No

Reviewer #2: Yes

4. Is the manuscript presented in an intelligible fashion and written in standard English?

Reviewer #1: Yes

Reviewer #2: Yes

5. Review Comments to the Author

Reviewer #1: Thank you for inviting me to review this paper. This manuscript is well-written, and the authors provide a very good introduction with relevant background information on the topic and rationale for their study. The methods are clear, and the results are well-presented. I have only a few minor edits and suggestions for the authors to consider.

Abstract, Materials and Methods – The authors could consider adding the word “in” before “2018-2019”.

Methods, Analysis section (top of p. 9) – I’d recommend adding the word “examine” before “differences”.

Results, Acupuncture – There were non-statistically significant differences on categories of race between VA and community acupuncture patients. Statistically significant differences were observed between groups on age and region of residence.

Limitations – The authors could consider moving this section to the end of the Discussion. I’d also recommend considering changing the word “observe” to “observed” (p. 10), and the word “do” to “did” and “be” to “have been”, and “can” to “could” and “are” to “were”, and “have not sought” to “did not seek” (p. 11, sentence starting with, “We also do not control for …”).

Discussion, p. 12 – I’d recommend the authors consider revising the sentence starting with, “While most studies comparing VA and non-VA …” to something like, “While most studies comparing VA and non-VA care examine inpatient and emergency care, prior studies of outpatient care have found that VA outpatient providers provide better or similar patient experiences to community outpatient providers.29,30”

Reviewer #2: Thank you for the invitation to evaluate “Patient-reported pain and physical health for acupuncture and chiropractic care delivered by Veterans Affairs versus community providers. The authors a report on quality improvement project to evaluate outcome variation (pain and physical function) for Veterans at the 18 VA Whole Health Flagship sites who received at least 4 visits of VA or-VA acupuncture or chiropractic care. The methods and statistical approach are appropriate for this investigation. The findings of this study add valuable preliminary insight to the differences in care outcomes when Veterans are rendered care at VA or community. I applaud the authors for their efforts.

As there are no line numbers in the proof, I will reference section and paragraph number when possible.

Several considerations for the authors:

INTRODUCTION:

Paragraph 2: Additional citation for on-station chiropractic services could be considered:

Halloran SM, Coleman BC, Kawecki T, Long CR, Goertz C, Lisi AJ. Characteristics and Practice Patterns of U.S. Veterans Health Administration Doctors of Chiropractic: A Cross-sectional Survey. J Manipulative Physiol Ther. 2021 Sep;44(7):535-545. doi: 10.1016/j.jmpt.2021.12.005.

Lisi AJ, Brandt CA. Trends in the Use and Characteristics of Chiropractic Services in the Department of Veterans Affairs. Journal of Manipulative and Physiological Therapeutics. 2016;39(5):381-386. doi:10.1016/j.jmpt.2016.04.005

Paragraph 3, sentence 1: “Comparing quality of care provided....” Is this citable? What is the supporting evidence?

Additional context for the development and employment of the workforce to deliver acupuncture and chiropractic is relevant as these services are recent additions to VHA – acupuncture more so than chiropractic. Please revise to enhance the introduction. I have included supporting citations related to this workforce:

Kligler B, Niemtzow RC, Drake DF, Ezeji-Okoye SC, Lee RA, Olson J, Reddy KP. The Current State of Integrative Medicine Within the U.S. Department of Veterans Affairs. Med Acupunct. 2018 Oct 1;30(5):230-234. doi: 10.1089/acu.2018.29087-rtl. Epub 2018 Oct 15.

Olson J, Kligler B. Society for Acupuncture Research Turning Point: Acupuncture in the Veterans Health Administration. J Altern Complement Med. 2021 Jul;27(7):527-530. doi: 10.1089/acm.2021.0194.

Olson JL. Licensed Acupuncturists Join the Veterans Health Administration. Med Acupunct. 2018 Oct 1;30(5):248-251. doi: 10.1089/acu.2018.1298. Epub 2018 Oct 15. PMID: 30377460; PMCID: PMC6205763.

Lisi AJ, Khorsan R, Smith MM, Mittman BS. Variations in the Implementation and Characteristics of Chiropractic Services in VA: Medical Care. 2014;52:S97-S104. doi:10.1097/MLR.0000000000000235

Lisi AJ, Goertz C, Lawrence DJ, Satyanarayana P. Characteristics of Veterans Health Administration chiropractors and chiropractic clinics. JRRD. 2009;46(8):997. doi:10.1682/JRRD.2009.01.0002

METHODS:

Was this manuscript reporting consistent with study design guidelines? For example, SQUIRE (quality improvement) or STROBE (cohort)?

Paragraph 2: Survey Sampling Approach:

Second sentence: For VA’s Whole Health demonstration project, please consider citing the initiative.

While a dose of 4 visits at least seems reasonable from a course of care where Veterans may be lost to follow-up or achieve outcome. Is there a source to support the rationale for dose selection?

RESULTS:

Page 9, Acupuncture: Race is noted as significant between VA and community acupuncture patients. In contrast, Table 1 fails to bold ‘race’ for acupuncture indicating in current form the difference is not significant. Please revise for continuity.

Regarding limitations, is there something unmeasured related to the 18 Whole Health Pilot VAs?

What was the representation of Veterans across the 18 medical centers? How do I know that I am on looking at the comparison of Veterans at a single site and not the suggested cross-section of the 18 VAs. Is there clarity to the distribution? Perhaps this was lost with the data de-linking step. If this is available, please consider reporting.

DISCUSSION

Paragraph 1 of the introduction frames interest in determining the advantages or disadvantages of referring to one care setting or another – VA vs. non-VA. Please consider revisions to the Discussion to draw on this notion. Considerations beyond the scope of this study, but worth commenting on:

1) Do on station services compared to non-VA services have increased likelihood of care coordination and utilization for other VA services thus enhancing use of VA care.

Thomas ER, Zeliadt SB, Coggeshall S, et al. Does Offering Battlefield Acupuncture Lead to Subsequent Use of Traditional Acupuncture? Medical Care. 2020;58:S108-S115.

Etingen, B., Smith, B.M., Zeliadt, S.B. et al. VHA Whole Health Services and Complementary and Integrative Health Therapies: a Gateway to Evidence-Based Mental Health Treatment. J GEN INTERN MED 38, 3144–3151 (2023).

2) Do divergent downstream care cascades occur between those Veterans who attended VA or non-VA care? Some of these are guideline discordant and commonly low-value health services: imaging, specialist referrals, and opioids prescriptions.

3) Does divergent health care utilization costs result from use of VA vs non-VA care?

4) Finally, overall cost episode and cost per visit is worth mentioning. Outcomes are similar at VA and non-VA after adjusting for demographics, visit count. However, what this does tell me is on-station VA care is more efficient in overall cost. Both policy makers, administrators, referring physicians, and patients should recognize the value of cost-effective care when having to choose where and when to ‘buy’ the ‘outcome’ when choosing VA or non-VA care.

Table 1: Consider including n for each ‘n’ in addition to ‘%’

6. PLOS authors have the option to publish the peer review history of their article (what does this mean?). If published, this will include your full peer review and any attached files.

Reviewer #1: No

Reviewer #2: **Yes: **Zachary Cupler

---

## [Editor Report · Decision Letter 1]

30 Apr 2024

Patient-Reported Pain and Physical Health for Acupuncture and Chiropractic Care Delivered by Veterans Affairs Versus Community Providers

PONE-D-24-02440R1

Dear Dr. O'Hanlon,

We’re pleased to inform you that your manuscript has been judged scientifically suitable for publication and will be formally accepted for publication once it meets all outstanding technical requirements.

Kind regards,

Jenny Wilkinson, PhD

Academic Editor

PLOS ONE

Additional Editor Comments (optional):

Thank you for your revisions and responses to reviewer comments. The only additional comment is that the word 'Legend' appears at the end of titles for Tables 1 and 3; this seems in error so should be deleted.
---

## [Editor Report · Acceptance letter]

3 May 2024

PONE-D-24-02440R1 

PLOS ONE

Dear Dr. O’Hanlon, 

I'm pleased to inform you that your manuscript has been deemed suitable for publication in PLOS ONE. Congratulations! Your manuscript is now being handed over to our production team.

Kind regards, 

on behalf of

Dr Jenny Wilkinson 

Academic Editor

PLOS ONE